# Credit Constraint, Interlinked Insurance and Credit Contract and Farmers' Adoption of Innovative Seeds-Field Experiment of the Loess Plateau

**Leshan Yu [1], Yan Song [2,3], Haixia Wu [4,\*] and Hengtong Shi [1,\*]**

[1] International Business School, Shaanxi Normal University, Xi'an 710119, China
[2] Sino-Danish College, University of Chinese Academy of Sciences, Beijing 101408, China
[3] School of Public Policy and Management, University of Chinese Academy of Sciences, Beijing 101408, China
[4] Institute of Agricultural Resources and Regional Planning, Chinese Academy of Agricultural Sciences, Beijing 100081, China
\* Correspondence: wuhaixia@caas.cn (H.W.); shihengtong@snnu.edu.cn (H.S.)

**Abstract:** The interlinked insurance and credit contract is an emerging model of agricultural insurance in China. However, the development of interlinked insurance and credit contract and farmers' demands for it are poorly understood. Based on the wheat farmers on the Loess Plateau in China, a field experiment is employed to obtain dynamic choice data from 415 farmers. We empirically analyzed the inhibitory effect of credit constraint on farmers' adoption behavior of innovative seeds and also explored the heterogeneity of farmers' innovative seeds adoption due to the availability of interlinked insurance and credit contract. The results illustrate that credit constraint can hinder farmers' innovative seeds adoption significantly, and interlinked insurance and credit contracts can encourage farmers to adopt innovative seeds effectively by dispersing natural risks and alleviating credit rationing. Further, constrained by low education levels in China's rural areas, providing interlinked insurance and credit contract to farmers is not beneficial to enhance farmers' innovative seeds adoption. In addition, farmers who are relatively poor may underestimate the benefits of innovative seeds at the beginning of planting, making their adoption behavior have some delayed effect. This research provides a new perspective for promoting the spread of innovative technology in rural areas.

**Keywords:** credit constraint; interlinked insurance and credit contract; technological adoption; innovative seeds; field experiment





## 1. Introduction

Climate change triggers the frequency and severity of natural disasters [1], which in turn will harm the agricultural sector and exacerbate the food crisis. A report jointly released in 2020 by the Food and Agriculture Organization of the United Nations, the International Fund for Agricultural Development, the World Food Program, and other agencies shows that 690 million people worldwide will go hungry as the new crown pneumonia epidemic, extreme weather, rising international energy prices, and geopolitical conflicts overlap. There is an obvious trend that the world is on the verge of the most serious food crisis in at least 50 years, and food crises will be more pronounced in developing countries, which are less resilient to natural disasters [2]. Therefore, improving farmers' resilience to disasters, promoting sustainable agricultural development, and coping with the global environment of uncertainty have become urgent issues worldwide.

Given the complex and diverse geographical and climatic conditions as well as the high risk and frequency of natural disasters, China's agricultural production is exposed to greater systemic risks [3,4]. According to China's 2020 National Economic and Social Development Statistical Bulletin, in 2020, China suffered the largest flood since

1998, resulting in a crop damage area of 19,957.7 thousand hectares, of which 2706.1 thousand hectares were extinguished, with a direct economic loss of CNY 370.15 billion. Moreover, rural households are characterized by insufficient resources and a high degree of concurrent industrialization, which hinders farmers' investments in productivity-enhancing technology [3,5]. Constrained by the long agricultural production cycles and lack of risk management tools [6], the agricultural industry's huge production losses are always difficult to disperse when natural disasters occur, which easily results in falling into the poverty trap [7]. Growing evidence indicates that the adoption of innovative agricultural technology is an important means for farmers to resist natural risks, increase farm income, and stabilize agricultural production [8–10], which is helpful in improving agricultural production efficiency, ensuring food security, and alleviating rural poverty [11–13].

China's agricultural science and technology have developed rapidly. After ten years of development, the contribution rate of agricultural science and technology progress exceeds 61%, which is an increase of 7% ("The 13th Five-Year Plan" China Agricultural and Rural Science and Technology Development Report). Among them, seeds are the most basic and important investment for agricultural production and an important carrier of agricultural science and technology [14]. Seed innovation is important for increasing agricultural production and income and ensuring national and even global food security. Since the enactment of the seed law in 2000, China has gradually embarked on an independent and comprehensive path of seed industry development. At present, China's superior seed coverage rate exceeds 96%, contributing to more than 45% of grain production. However, foreign dependence on crop seeds remains high. According to the Chinese Seed Industry Development Report 2021, China's seed trade deficit was as high as USD 230 million in 2019, and there is still a large gap between the seed industry's independent innovation and that of developed countries. However, farmers usually encounter the dualistic economic structure [15], risk allocation, and price allocation in China [16,17]. Due to credit constraints, farmers' adoption of new agricultural technology is severely inhibited [11,16–18], which further hinders the improvement of income and welfare levels [19,20]. It is obvious that for developing countries such as China, there is a long way to go to accelerate the development of the modern seed industry.

In this circumstance, purchasing agricultural insurance became an efficient way to resist agricultural risks and economic losses [21–24]. Existing studies suggest that agricultural insurance supports farmers' technology investment activities in two main ways: on the one hand, by increasing farmers' risk tolerance and thereby changing their risk coping strategies [6,25]. On the other hand, by alleviating farmers' financial constraints to boost their agricultural investment [26,27]. The Chinese government has attached great importance to the promotion and improvement of agricultural insurance since 2004. What is of great concern is that the keyword "insurance" is mentioned 11 times in the No. 1 document of the Central Government in 2022. Agricultural insurance in China has come a long way in nearly 20 years, yet the effectiveness of agricultural insurance in promoting farmers' adoption of innovative agricultural technology is not satisfactory [28,29]. Furthermore, the insurance companies lack the incentive to innovate and update targeted insurance products, which stems from the policy-based agricultural insurance system. As it continues to evolve, such a system gradually highlights institutional weaknesses (e.g., adverse selection, moral hazard, over-reliance on government subsidies, etc.) [30]. Moreover, due to the asymmetry of information and lack of trust, the demand for and acceptance of agricultural insurance are still low in China [31].

Therefore, more and more studies emphasize that interlinked insurance and credit contract is a valuable tool that can address chronic poverty caused by insurance and credit market failures in low-income countries efficiently [11]. This is because, compared to traditional agricultural insurance, the linkage between insurance and credit markets can alleviate financial constraints on farmers effectively, transfer agricultural systemic risks, and promote farmers' adoption behaviors of agricultural technology [32,33]. Moreover, a

large body of studies shows that interlinked insurance and credit contracts can transfer agricultural production risks, reduce the lending risks of financial institutions, expand farmers' demand for credit, and weaken the risks arising from farmers' self-selection. In other words, the contract is a "stabilizer" that can facilitate the effective adoption of innovative technology by farmers [34–36]. However, some scholars hold the opposite view and argue that interlinked insurance and credit contract do not promote farmers' adoption of innovative technology. Farmers who purchase only an insurance contract are more receptive to technology use than farmers who purchase an interlinked insurance and credit contract. This can be explained in terms of the cost of loan defaults, where smaller default penalties motivate farmers to adopt higher levels of technology [19,37]. At the same time, farmers' agricultural technology adoption behaviors are heterogeneous due to different social environments, cultural backgrounds, and agricultural patterns. In addition, factors such as model differences, type of technology, and within-sample variability can also make farmers' technology adoption behavior variable [38,39].

In light of the preceding analysis, the aim of this paper is to verify the inhibitory effect of credit constraints on farmers' adoption of innovative seed technology through data obtained from a field experiment. Then we examine farmers' willingness to adopt innovative seeds in the presence or absence of interlinked insurance and credit contract. This study specifically addresses the adoption of traditional and innovative wheat seeds among wheat farmers in the Shaanxi and Shanxi provinces of China. The following three points summarize this paper's marginal contributions: First, we examine the moderating effect of credit constraint on the inhibition of farmers' adoption of innovative seeds, given the mechanism by which interlinked insurance and credit contract promote farmers' adoption of innovative seeds. Second, we analyze the impediments to the demand for interlinked insurance and credit contract in the current stage of China's agricultural development, which provide a theoretical reference for the further development of interlinked insurance and credit contract. Third, we simulate the real situation with a field experiment, allowing farmers to make dynamic choices based on their understanding of the operation mechanism of interlinked insurance and credit contract, overcoming the disadvantage that questionnaire surveys can only obtain static time-point indicators.

The rest of the paper is structured as follows: Section 2 is theoretical analysis, Section 3 focuses on sample selection and field economics experimental design, Section 4 is empirical analysis, Section 5 is the discussion, and Section 6 concludes the full paper and makes policy recommendations.

## 2. Theoretical Analysis

### 2.1. Credit Constraint and Farmers' Adoption of Innovative Seeds

The early theories of credit availability and credit rationing laid the foundation for the development of credit constraint theory. The role of credit constraints in discouraging farmers from investing in innovative technology is widely recognized by academics. For example, Carter and Olinto [40,41] indicate that farmers with liquidity constraints will invest less when the credit supply is inadequate. Shiferaw et al. [42] also provided empirical support for the finding that credit constraint hinders farmers' adoption of innovative agricultural technology. Tesfaye et al. [43] concluded that smallholder farmers tend to operate below the production possibility frontier because of financing constraints that prevent them from adopting more efficient and labor-saving irrigation technology. Therefore, improving smallholder farmers' access to credit is necessary. However, with the gradual advancement of research, more and more researchers find that in addition to credit, constraints arise from credit rationing, farmers' own risk aversion, and cognitive biases also contribute to credit constraint [44,45].

For supply-based credit constraints, financial institutions in rural areas are constrained by high business costs and risks, as well as information mismatches, and engage in interest rate regulation to reduce credit supply and implement credit rationing on a property basis. Then, it is difficult to satisfy farmers' loan requirements, and agricultural technology with

high investment is restricted. On the other hand, farmers, as the main demanders of credit, may encounter credit repression or credit substitution due to risk aversion and cognitive preferences. Therefore, farmers' motivation to take out loans may be reduced because of the fear that the credit amount will not satisfy their needs or the fear of losing collateral [46–48], and the adoption of agricultural technology will be limited as a result. Therefore, we propose hypothesis 1 in this paper.

**Hypothesis 1.** *Credit constraint inhibits farmers' adoption of innovative seeds.*

### 2.2. Interlinked Insurance and Credit Contract and Farmers' Adoption of Innovative Seeds

Most of the existing literature has concluded that insurance can produce the same effect as credit collateral to a certain extent by spreading the farmer's credit risk [21,49]. The "interlinked insurance and credit contract" approach not only helps credit institutions transfer their own lending risks but also helps farmers obtain the financial credit support needed for the development of agriculture [32,49,50], thus encouraging farmers to adopt innovative technology [37,51]. Rural financial institutions in China serve more small-scale farmers, who have a dispersed spatial distribution and significant information asymmetry issues [23]. Interlinked insurance and credit contract provides insurance for farmers, which can help them solve the problem of insufficient credit effectively. It can also definitely reduce the moral hazard problem in the contract performance process. Therefore, we believe that the cooperative approach of interlinked insurance and credit contract can alleviate farmers' credit constraints effectively and promote their adoption of innovative technology. Therefore, we propose hypothesis 2 in this paper.

**Hypothesis 2.** *Interlinked insurance and credit contract can promote farmers' adoption behavior of innovative seeds effectively.*

### 2.3. Moderating Effects of Interlinked Insurance and Credit Contract

The credit and insurance interconnection paradigm was formally proposed in China in the 2009 Central Government Document No. 1, which stated that interlinked insurance and credit contract is one of the most important measures for dispersing agricultural risks [10,12,45,52]. The easing of credit constraints provides not only financial support for farmers' investments in innovative seeds but also supplies large amounts of capital inputs in agricultural production for farmers. A number of papers indicate that due to the risk dispersion of interlinked insurance and credit contract, credit institutions are always willing to enlarge the credit amount to farmers [19,53]. Therefore, it is acknowledged that the impact of credit constraints on farmers' adoption behavior of innovative seed technology may considerably depend on the participation of interlinked insurance and credit contract products. That is, if farmers choose to purchase the interlinked insurance and credit contract, their constraint on risk and credit will be alleviated, and moral hazard in the process of financial contract performance will be reduced. Therefore, we propose hypothesis 3 (Figure 1) in this paper.

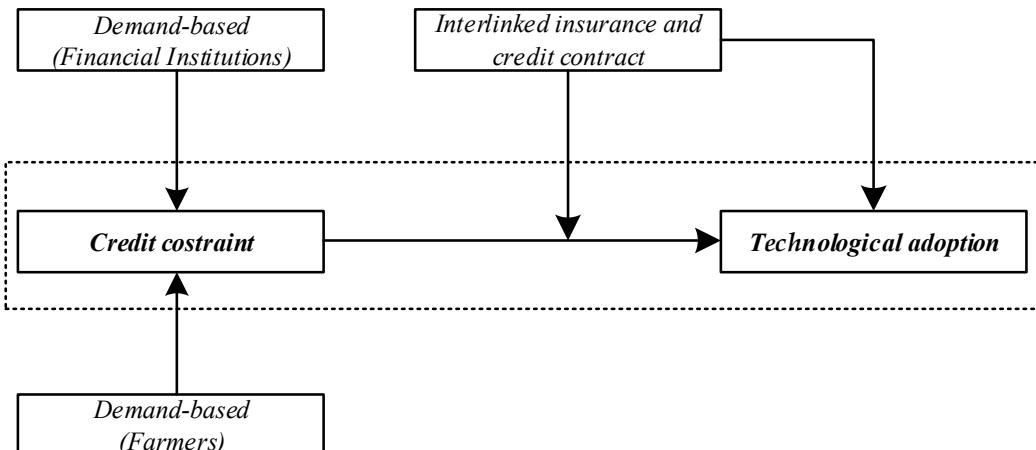

**Figure 1.** Mechanism analysis.

**Hypothesis 3.** *The interlinked insurance and credit contract can alleviate the inhibitory effect of credit constraint on the framers' adoption of innovative seeds effectively.*

## 3. Experiment Design

### 3.1. Experiment Set and Distribution of Samples

The data used in this study come from the survey and field experiment that we implemented in July and August of 2021 (Appendix C). By using the multi-stage stratified sampling method, we selected a total of 415 wheat farmers in the Shaanxi and Shanxi provinces of China as sample farmers (Figure 2). We first selected the sample wheat farmers from four counties randomly. In Shaanxi Province, this includes Heyang and Yongshou counties, while in Shanxi Province, it includes Yaodu and Pinglu counties. Second, considering the level of economic development and geographical location, five towns were selected in each sampling county (Table 1). Finally, we obtained the list of villagers from the local village committees and selected the sampling farmers according to the principles of 2, 4, and 10 distances for villages with less than 50, 51–100, and more than 100 households, respectively. Figure 3 illustrates the sampling procedure.

The principles of area selection are based on two aspects:

The first is crop cultivation systems. Wheat is the main food crop in the Loess Plateau region of China, where agricultural production conditions and climatic conditions vary greatly. In China, the wheat cultivation system is complex, with three categories: winter wheat once a year, two-crop winter wheat and summer corn once a year, and three-crop winter wheat and summer corn (other grains)—spring corn twice a year. In our study areas, Yongshou County is a one-crop winter wheat planting area; Heyang County and Yaodu District are one-crop and two-crop planting areas; and Pinglu County is a mixture of two-crop a year and three-crop twice a year. The above areas are important wheat-producing areas in northern China. It is of great significance to study the adoption of new wheat technology in these areas to guarantee national food security.

Second, the degree of technological development in wheat cultivation. Shanxi Province and north-central Shaanxi Province in China are part of the Loess Plateau region. Due to low precipitation and dry weather, soil erosion and loss of fertility in these areas are severe, threatening the quality of wheat agricultural development and sustainable agricultural development in these regions. Heyang and Yongshou counties in Shaanxi Province are identified as dryland wheat integration trial areas by the Department of Agriculture and Rural Affairs of Shaanxi Province in the 2017 Shaanxi Wheat Trial Implementation Plan. Additionally, Heyang County in Shaanxi Province was listed as a national agricultural science and technology modernization pioneer county in 2021 by the Ministry of Agriculture and Rural Affairs of the People's Republic of China. To encourage the introduction of science and technology into rural areas, Shaanxi Province introduced the "Agricultural

Technology Promotion Achievement Award". Pinglu County in Shanxi Province built 16,000 mu of organic wheat dry farming and water-saving agriculture demonstration park in 2019 and formulated the "Implementation Plan for the Construction of Organic Dry Farming Organic Wheat Demonstration Area in Pinglu County" to guide the sowing of good varieties and improve the quality of wheat. Moreover, based on the 14th Five-Year Plan, Yao Du District launched the organic dry farming wheat cultivation advice in 2021, which provides detailed guidance on dryland wheat technology selection and variety selection to further demonstrate the effectiveness of organic dry farming development.

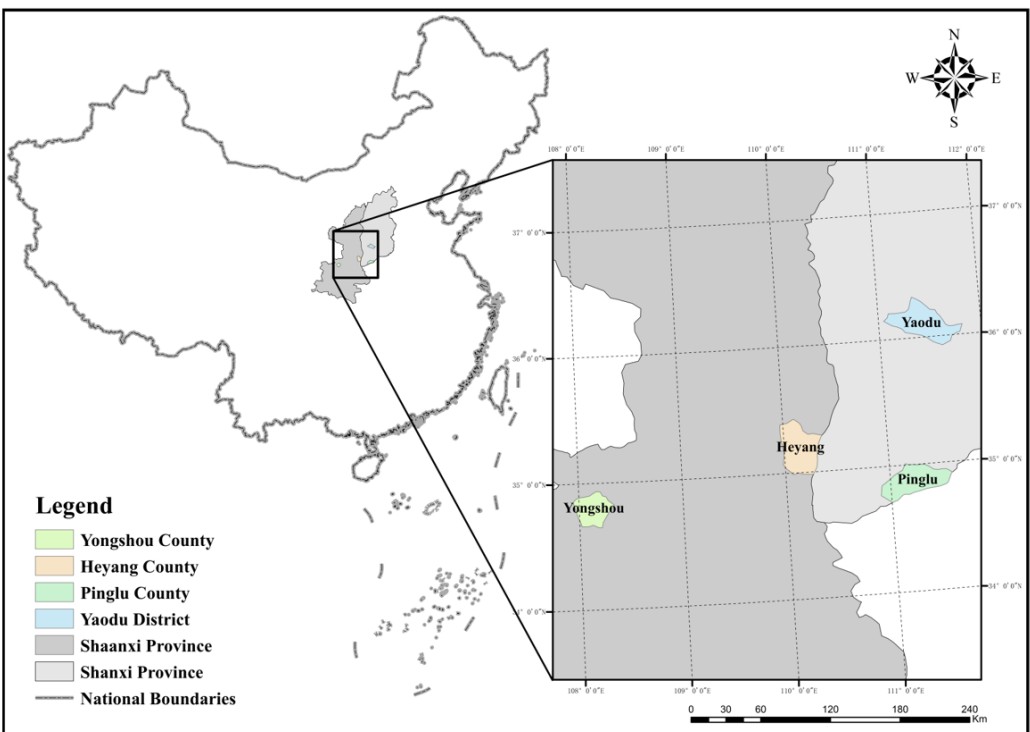

**Figure 2.** Sample locations.

**Table 1.** Sample distribution.

| Province | Sample Cities (Counties) | Sample Cities (Counties) | Number of Samples | Percentage |
|---|---|---|---|---|
| Shaanxi Province | Yongshou County | Changning Town, Ganjing Town, Quzi Town, Dian Tou Town, Jianjun Town | 79 | 19.04% |
| | Heyang County | Wangcun Town, Lujing Town, Heichi Town, Xinchi Town, Fang Town | 178 | 42.89% |
| Shanxi Province | Yaodu District | Jindian Town, Tumen Town, Qiaoli Town, Wucun Town, Xiandi Town | 75 | 18.07% |
| | Pinglu County | Shengrenjian Town, Zhangdian Town, Sanmen Town, Changle Town, Podi Town | 83 | 20.00% |

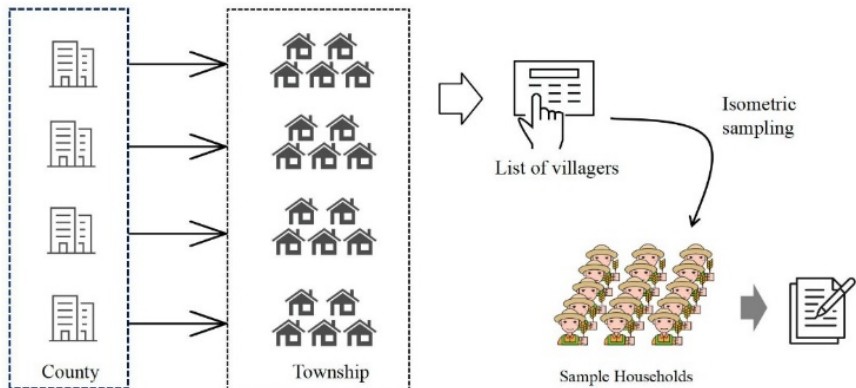

**Figure 3.** Illustration of stratified sampling.

Therefore, it is believed that the above sites can represent farmers' adoption behavior of innovative seeds in the Loess Plateau region of China.

*3.2. Experimental Design*

3.2.1. Research Methodology

Both theory and practice show that the interlinked insurance and credit contract has grown in China in recent years, but their further development is hampered by small-scale and regional heterogeneity. Based on the survey, we learned that the type of crop seeds is the most important factor affecting the yield, and farmers also attach great importance to crop seed varieties. As a result, we selected innovative seeds to represent new technology.

It is well known that agricultural production is continuous and farmers' technology selection behavior is dynamic, so it would be difficult to examine the impact of interlinked insurance and credit contract on farmers' adoption of innovative seeds through a traditional questionnaire survey. Consequently, in order to reduce survey bias and ensure a more accurate estimation of results, a field experiment was adopted to collect data concerning farmers' preferences for interlinked insurance and credit contract and innovative seeds adoption [54]. Firstly, we selected a sample of subjects from an overall population randomly and divided them into a control group and a treatment group; the sample from the treatment group was treated experimentally. Since subjects are randomly divided into two groups, the treatment group is completely independent of individual characteristics and other factors that might affect the experimental results, which avoids the problem of omitted variable bias or endogenous variable bias commonly [55], so the experimental process of the field experiment is close to the real world and could make tests of causal relationships between variables more direct and convenient.

3.2.2. Research Methodology

1. **The setting of groups.** (1) Control group. Farmers in this group choose between traditional wheat seeds that do not require credit and innovative wheat seeds that do; (2) Treatment group. Farmers in this group make decisions between traditional wheat seeds that are not financially constrained and innovative wheat seeds that provide interlinked insurance and credit contract.

2. **The setting of production conditions.** Referring to the study of Tang et al. [6], we assume that the farmers owe CNY 4200 in terms of capital and 10 Mu cultivated land at the beginning. In the first year, farmers are required to make a choice of wheat seeds between No. 0 seeds and No. 1 seeds. The differences in seeds, production inputs, and income under different weather conditions for these two kinds of seeds are shown in Table 2. If farmers choose No. 0 seeds, their own capital is enough to satisfy the demand for production; if they choose No. 1 seeds, farmers need to apply for a loan of CNY 1800 and submit collateral worth CNY 1800 (we assume the farmers can afford it). At the end of the year, if the weather is suitable for wheat growth, the farmers

who planted No. 1 seeds will obtain the collateral back after repaying the loan; if the weather is bad, the farmers planting No. 1 seeds are unable to repay the loan, and the bank will confiscate the collateral, so farmers will lose CNY 1800. Farmers who do not go bankrupt in the first round of experiments will continue with the second round of experiments, and the final payoff in this experiment will be determined by the remaining funds in each farmer's hand at the end of the last year.

3.  **The setting of weather conditions.** Considering that the weather conditions in the previous year may affect planting plans for the next year, we simplify the weather conditions in the areas into two categories: disaster (bad weather) and normal (good weather). Farmers randomly draw a card from the black box containing four red cards and two black cards; the black card represents bad weather, while the red card represents good weather. According to the local meteorological information, we can know that the incidence of disaster weather is one-third. Farmers are required to determine the natural conditions they face by drawing cards randomly, ensuring that they do not know the weather conditions of the year. Moreover, the loss will be 1.55 times higher than that of normal seeds if bad weather occurs. The main parameters are presented in Table 2.

**Table 2.** Experimental parameter settings comparison of traditional and innovative seeds.

| Experimental Group | Seed Variety | Production Input | Loan | Premium | Planting Income (Good) | Balance (Good) | Planting Income (Bad) | Balance (Bad) |
|---|---|---|---|---|---|---|---|---|
| Control group | No. 0 seeds | 4000 | 0 | 0 | 7000 | 7200 | 0 | 200 |
| | No. 1 seeds | 6000 | 1800 | 0 | 12,000 | 10,200 | 0 | −1800 |
| Treatment group | No. 0 seeds | 4000 | 0 | 0 | 7000 | 7200 | 0 | 200 |
| | No. 1 seeds | 6000 | 2000 | 200 | 12000 | 10,000 | 0 | 0 |

4   **The experimental process (Appendix B).** Before the experiment, we explained to farmers, in text (the experimental instruction manual), the details of the experimental situation, the main parameter, the task arrangements of the experiment, and the relevant requirements. Additionally, the tester continued to demonstrate the experimental process until the farmers fully understand the entire experimental content. To avoid communications between farmers, four farmers were taken by each tester and separated by baffles to ensure the independence of choice. If farmers in the control group choose No. 0 seeds, their own capital is enough to maintain agricultural production. If the weather is suitable for planting (good weather), they earn CNY 7000 and have a balance of CNY 7200. If the weather is not suitable (bad weather), they have no income and a balance of CNY 200. If farmers choose No. 1 seeds, they need to apply for a credit of CNY 1800. Moreover, they are required to provide the banks with equally valued collateral. At the end of the year, they receive the collateral back and obtain a balance of CNY 10,200 if the weather is good. However, if bad weather occurs, farmers who choose No. 1 seeds lose the collateral worth CNY 1800. The loss to farmers is CNY 1800. Since farmers experiencing bad weather in the first year would fall into bankruptcy, we added an exit option in the second year (i.e., withdrawing from agricultural production). The experiment was conducted one more time while other scenarios were the same as in the first year.

In the treatment group, a game was implemented to help farmers in the experimental group understand the difference between a traditional insurance contract and an interlinked insurance and credit contract. Next, the farmers were asked to answer some simple test questions to see if they fully understood the contract. If the farmer still did not understand the experiment, the tester explained it in greater detail until they did. In terms of the treatment group, the basic conditions are the same as those in the control group, and the only difference is that the farmers who chose No. 1

seeds were required to purchase an interlinked insurance and credit contract with a premium of CNY 200. Therefore, farmers needed to borrow CNY 2000 from the bank and also provided collateral of CNY 2000. At that point, if the weather was suitable for planting (good weather), they received a satisfactory output. However, if bad weather occurs, the insurance company pays the bank first, and the farmers receive the collateral back. They will have no income and a balance of CNY 0.

5　**The measurement of farmers' risk attitude.** Risk attitude is a major force influencing farmers' decisions and an important factor in the adoption of innovative seeds by farmers due to credit constraints and interlinked insurance and credit contracts [32,56]. In order to obtain accurate information concerning farmers' risk attitudes, we measured farmers' risk attitudes through a field experiment. First, farmers were informed that there are three black cards and three red cards in a transparent bag, and the rewards for drawing the black and red cards are shown in Table 3. Then, farmers are required to make their choices. Only the farmers who chose Plan B1 proceeded with the game, as it could help us improve the accuracy of formal experiments and reduce invalid samples.

**Table 3.** Test Games.

| Risk Options | Plan A1 | | Plan B1 | |
|---|---|---|---|---|
| | Red card | Black card | Red card | Black card |
| | 15 | 20 | 16 | 21 |

Ten sets of formal tests were set up (see Table 4), each of which includes both low-risk and high-risk reward options. Farmers select either reward Plan A2 or B2 from each set, of which Plan A2 implies low risk while Plan B2 implies high risk. In the experiment, we established two premises: first, farmers were explicitly informed that there were three black cards and three red cards in the bag. Second, there are six cards in total, but the colors are unknown to farmers. The two settings above are used to measure the risk attitude indices with definite probability and ambiguous probability, respectively, according to the farmers' choice (Equations (1) and (2)).

$$Risk_d \ = \ \frac{N - B2_{nd}}{N} \tag{1}$$

$$Risk_d \ = \ \frac{N - B2_{nd}}{N} \tag{2}$$

where $Risk_d$ and $Risk_f$ indicate the risk level under definite probability and the risk level under ambiguous probability, respectively; $N$ is the total number of experiments; $B2_{nd}$ is the number of times that who chose reward Plan B2 with definite probability, while $B2_{nf}$ is the number of times that who choice reward Plan B2 with ambiguous probability. The risk attitude level has a value range of [0,1], where 1 indicates that the farmer is extremely risk averse and 0 indicates that the farmer extremely prefers risks.

Individual characteristics of the household head, family characteristics, participation in technological training, and other external information are obtained from the questionnaire (Appendix A). The whole experiment lasts about 80 min, and the farmers will be paid accordingly (around CNY 60, which is equal to 1/1000 of the annual budget in the experiment) when they complete the experiment.

**Table 4.** Experimental protocol.

| Options | Plan A2 | | Plan B2 | |
|---|---|---|---|---|
| | Red Card | Black Card | Red Card | Black Card |
| 1 | 20 | 20 | 22 | 18 |
| 2 | 20 | 20 | 23 | 17 |
| 3 | 20 | 20 | 25 | 15 |
| 4 | 20 | 20 | 35 | 15 |
| 5 | 20 | 20 | 37 | 13 |
| 6 | 20 | 20 | 40 | 10 |
| 7 | 20 | 20 | 52 | 8 |
| 8 | 20 | 20 | 54 | 6 |
| 9 | 20 | 20 | 56 | 4 |
| 10 | 20 | 20 | 60 | 0 |

*3.3. Control Variables*

3.3.1. Selection of Control Variables

In order to avoid the impact of other factors on the results, we control the characteristics of the household head, the characteristics of the household, and the support of the government [43,57].

3.3.2. Descriptive Statistics of the Variables

The definitions and descriptive statistics of variables are shown in Table 5. The average age of the farmers is 56.93 years, and the average years of education are 7.51 years; most of the farmers have not completed junior high school or higher education ($\geq$9 years). In terms of the change from wheat cultivation area to cropland area, the proportion is 68% on average, and wheat is the main crop in the sample area, which is also in line with the field experiment. The distance of farmers from the nearest financial institutions, such as rural credit institutions, is 5.3 miles on average, which may affect relevant services, information transmission, etc. In the past three years, the average number of times farmers attended training on wheat growing techniques is less than 1, which indicates the low willingness of farmers to participate in agricultural technology training.

*3.4. Model Setting*

According to the field experiment, the dependent variable in the first round is the binary choice between innovative wheat seeds and traditional wheat seeds. In order to test the effect of credit constraint on the framer's adoption of innovative seeds, we first constructed a binary *Probit* model as follows:

$$Probit(Choice = 1|Credit, x) = \varphi(Credit\beta + x\theta) \tag{3}$$

where *Choice* indicates the farmer's choice of wheat seeds; if the farmer chooses innovative wheat seeds, *Choice* = 1; otherwise, we assigned the value of 0. *Credit* indicates credit constraint, which is used to examine the effect of credit constraint on a farmer's adoption behavior of innovative wheat seeds, we assigned the value of 1 if the farmer belonged to the treatment group; otherwise, we assigned the value of 0. $x$ is the vector of control variables; $\beta$ and $\theta$ are regression coefficient estimates; and $\varphi(\cdot)$ is a normally distributed probability function.

**Table 5.** Variable definition and descriptive statistics.

| Variables | Meaning and Assignment of Variables | Mean | S.D. |
|---|---|---|---|
| Choice | Choice1: The technology selection in the first round of experimentTraditional seeds = 0, Innovative seeds = 1 | 0.52 | 0.50 |
| | Choice2: The technology selection in the second round of experimentTraditional seeds = 0, innovative seeds = 1 | 0.54 | 0.49 |
| Credit constraint | If CNY 50,000 is needed for production turnaround, how easy is it to borrow? (1 = very difficult; 2 = a little bit difficulty; 3 = okay; 4 = easy; 5 = very easy) | 3.43 | 1.22 |
| Interlinked insurance and credit contract | If the farmer belongs to treatment group, then assign thevalue of 1; otherwise, then assigned 0 | 0.46 | 0.49 |
| Age | The actual age of the respondent, Unit: year | 56.93 | 9.49 |
| Education | Years of education of respondent, Unit: year | 7.51 | 2.89 |
| Leader | Is the head of the household a village official?1 = Yes; 0 = No | 0.16 | 0.36 |
| Income | Total income of the sample households in the last year., unit: Yuan | 1.42 | 5.53 |
| Number | Number of plots planted with wheat, Unit: block | 4.44 | 29.38 |
| Labor | Number of family agricultural laborers | 2.05 | 0.90 |
| Ratio | Ratio of wheat cultivation area to cultivated area (%) | 0.68 | 0.29 |
| Financial | Are there any family members or relatives working in financial institutions? 1 = Yes; 0 = No | 0.04 | 0.19 |
| Insurance | Did your household take out insurance for growing wheat last year? (1 = yes; 0 = no) | 0.51 | 0.50 |
| Risk | Measured by the Farmers' Risk Attitude Test | 0.32 | 0.35 |
| Training | Number of times respondents attended training on wheat growing techniques in the past year | 0.69 | 2.42 |
| Information | Does the village provide technological information services for defense against weather hazards? 1 = Yes; 0 = No | 0.52 | 0.50 |
| Distance | How far is your home from the nearest financial institution, such as a rural credit union? Unit: mile | 5.30 | 3.73 |
| Perception | How do you think the local precipitation in the last 5 years? 1 = significantly decreased; 2 = somewhat decreased; 3 = not significantly changed; 4 = somewhat increased; 5 = significantly increased | 2.81 | 0.93 |
| Climate | Do you think the local climate has been warming in the last 30 years? 1 = very disagree; 2 = disagree; 3 = neutral; 4 = agree; 5 = very agree | 4.23 | 0.88 |
| Province | 0 = Shanxi, 1 = Shanxi | 0.62 | 0.49 |

According to the design of the field experiment, farmers who chose innovative wheat seeds in the first round of the experiment with bad weather dropped out of farm production due to bankruptcy, so their choice in the second round of the experiment became multiple choices. Therefore, we further examined the effect of credit constraint on farmers' adoption of innovative wheat seeds using the *MultipleProbit* model, which is set up as follows:

$$Probit(choice = j \mid x_i) = Probit\{\varepsilon_{ik} - \varepsilon_{ij} \leq (x_{ij}\text{-}x_{ij})\,\beta\} \tag{4}$$

where *Choice* indicates the farmer's choice of wheat seeds; if the farmer chooses innovative wheat seeds, *Choice* = 1; otherwise, we assign the value of 0. Both *j* and *k* indicates the farmer's technology choice options and $x_i$ is the explanatory variable.

## 4. Empirical Results

### 4.1. Baseline Regression

#### 4.1.1. First-Round Experimental Regression Results

Table 6 (the first two columns) reports the effects of credit constraints on farmers' adoption of innovative seeds in the first round of experiments. The regression model indicates that credit constraint has a significantly negative impact on farmers' innovative seeds choices at the level of 1%, which is consistent with the finding of Tesfaye et al. [43]. Hypothesis 1 proposed in this paper was tested; that is, credit constraint inhibits farmers' adoption of innovative seeds. Moreover, from the marginal effect regression results, we can see that the coefficient of credit constraint is −0.1932, which implies that when credit

constraint increases by 1 unit, the probability of adopting innovative seeds by farmers will decrease by 19%. Concurrently, interlinked insurance and credit contract has a significant positive effect on farmers' adoption of innovative seeds, and its coefficient is significantly positive at the level of 5%, implying that interlinked insurance and credit contract can promote farmers' adoption of innovative seeds, and Hypothesis 2 proposed in this paper is verified. According to the marginal effect regression results, for each 1% increase in the probability that a farmer purchases interlinked insurance and credit contracts, the probability that a farmer adopts innovative seeds will increase by 0.23%. This is consistent with the findings of Carter et al. [51] and Farrin et al. [37]. On the one hand, interlinked insurance and credit contracts can reduce farmers' credit rationing and address financial constraints. On the other hand, it makes farmers' technology investment risks to be mitigated.

Besides the effect of credit constraint and the "interlinked insurance and credit contract", the results in Table 6 reveal that among the individual factors, both years of education and the variable of leader have a positive effect on the farmers' adoption of innovative seeds at the level of 10% significantly, which indicates that the higher the education level, the better the farmer's understanding and ability to understand new technology [10]. As we learned in the questionnaire research, farmers who are village cadres have more opportunities to participate in relevant training and have more resources than ordinary villagers, which results in a more positive attitude toward the adoption of innovative seeds. Among the household factors, the coefficient of household income is significantly positive, indicating that the higher the household income, the more it can promote the farmers' adoption of innovative seeds. Household income is an important indicator of production and living conditions, and farmers with high levels of household income tend to have higher levels of part-time employment. At the same time, farming has higher opportunity costs [50]. Due to the high risk and cost of innovative seeds inputs, well-financed households not only have a higher willingness to replace traditional seeds with innovative seeds but also have the ability to afford the higher costs of innovative technology. The variable of risk preference is significantly positive at the level of 5%, which indicates that farmers with a stronger risk preference are more likely to be inclined to adopt high-risk, high-reward innovative seeds. This is in line with the study of Giné et al. [58], which concluded that the stronger the risk perception of individuals, the more inclined they are to take measures such as purchasing insurance to avoid risk, and they are more concerned with benefits than risks and therefore prefer new technology with high risks and high rewards [6]. In general, farmers who participate in technology training have a higher level of knowledge about innovative seeds and awareness of the economic benefits. They can learn more about the potential of the adopted seeds, thus reducing the risks and uncertainties associated with technology for farmers and promoting the adoption of innovative seed technology. However, the regression results show that "How many times have you attended training on wheat growing technologies in the last year?" is significantly negative, which is not consistent with expectations. We learned in our field experiment that most farmers may not be able to properly assess the potential impact of technology training and that they participate in training less than once on average. "Does the village provide technological information services for defending against meteorological disasters?" is significantly positive at the level of 5%, indicating that meteorological information services can alleviate information asymmetry and thus promote farmers' adoption of innovative seeds. Among the regional variables, "Distance to the nearest financial institution" has a significant negative effect on the farmers' adoption of innovative seeds. The closer the distance is to the financial institution, the lower the cost of information and time for farmers to obtain relevant credit policies, which is more conducive to farmers' innovative seeds adoption behavior [59,60].

**Table 6.** Regression results of probit model for two rounds of experiments.

| Variables | First Round of Experiments | | Second Round of Experiments | |
|---|---|---|---|---|
| | **Probit Model** | **Marginal Effect** | **Probit Model** | **Marginal Effect** |
| Credit constraint | −0.4851 *** | −0.1932 *** | −0.6038 *** | −0.2317 *** |
| | (0.07) | (0.02) | (0.08) | (0.03) |
| Interlinked insurance and credit contract | 0.6047 ** | 0.2369 *** | 0.5841 ** | 0.2197 ** |
| | (0.15) | (0.06) | (0.16) | (0.06) |
| Age | −0.0015 | −0.0005 | −0.0106 | −0.0041 |
| | (0.00) | (0.00) | (0.01) | (0.00) |
| Education | 0.0434 * | 0.0173 * | 0.0487 * | 0.0187 * |
| | (0.03) | (0.01) | (0.03) | (0.01) |
| Leader | 0.3557 * | 0.1391 * | 0.1707 | 0.0642 |
| | (0.21) | (0.08) | (0.22) | (0.08) |
| Income | 0.1392 ** | 0.0555 ** | 0.3721 ** | 0.1427 *** |
| | (0.07) | (0.03) | (0.11) | (0.04) |
| Number | −0.0369 | −0.0147 | −0.0328 | −0.0126 |
| | (0.02) | (0.01) | (0.03) | (0.01) |
| Labor | −0.0363 | −0.0145 | 0.0350 | 0.0134 |
| | (0.08) | (0.03) | (0.08) | (0.03) |
| Ratio | 0.4595 * | 0.1831 * | 0.3541 | 0.1358 |
| | (0.25) | (0.10) | (0.26) | (0.10) |
| Financial | −0.4997 | −0.1942 | −0.6386 * | −0.2505 * |
| | (0.39) | (0.14) | (0.39) | (0.15) |
| Insurance | −0.0115 | −0.0046 | 0.0277 | 0.0106 |
| | (0.16) | (0.07) | (0.17) | (0.07) |
| Risk | 0.5456 ** | 0.2174 ** | 0.4929 ** | 0.1890 ** |
| | (0.20) | (0.08) | (0.21) | (0.08) |
| Training | −0.1307 ** | −0.0521 ** | −0.0474 | −0.0182 |
| | (0.05) | (0.02) | (0.04) | (0.02) |
| Information | 0.4575 ** | 0.1808 ** | 0.3464 ** | 0.1325 ** |
| | (0.16) | (0.06) | (0.17) | (0.06) |
| Distance | −0.0466 ** | −0.0186 ** | −0.0334 | −0.0128 |
| | (0.02) | (0.01) | (0.02) | (0.01) |
| Perception | 0.0055 | −0.0048 | −0.1308 * | −0.0501 * |
| | (0.08) | (0.03) | (0.08) | (0.03) |
| Climate | 0.0534 | 0.0212 | −0.0231 | −0.0088 |
| | (0.09) | (0.04) | (0.09) | (0.04) |
| Distance | 0.1955 | 0.0778 | 0.2111 | 0.0814 |
| | (0.18) | (0.07) | (0.19) | (0.08) |
| Weather | | | −0.1236 | −0.0471 |
| | | | (0.16) | (0.06) |
| Constant | 0.0354 | | 1.3755 * | |
| | (0.85) | | (0.88) | |
| LR chi2 | 132.85 | | 165.10 | |
| Prob > Chi2 | 0.0000 | | 0.0000 | |
| Log likelihood | −220.96063 | | −203.4569 | |
| Pseudo R$^2$ | 0.2311 | | 0.2886 | |
| Number of samples | 415 | | 415 | |

Note: ***, **, * indicate significance at the 1%, 5%, and 10% statistical levels, respectively, and standard errors are in parentheses.

### 4.1.2. Regression Results of the Second Round of Experiments

Considering the continuity of agricultural production, the behavior of farmers' adoption of innovative seeds technology in the second year may be influenced by the weather in the first year, so we take the weather condition encountered in the first round of the experiment as a control variable in the second round. According to the regression results (Table 6), the variable credit constraint is significantly negative at the level of 1%, consistent with the results in the first round of experiments, and hypothesis 1 of this paper is tested again. The

coefficient of interlinked insurance and credit contract is significantly positive at the level of 5%, and interlinked insurance and credit contracts could promote farmers' adoption of innovative wheat seeds, and hypothesis 2 proposed in this paper is also tested again.

Among the control variables, unlike in the first round, household income is significantly positive at the level of 5%, and its marginal effect is larger than the marginal effect in the first round. The effect of income on innovative seed adoption is more pronounced in the second year, indicating that the farmers are more rational. As the experiment progressed, farmers understood the experiment more deeply, and their choice of innovative seeds required more household resources to cope with the occurrence of agricultural production risks. We can see that the weather condition in the first year is insignificant, which is different from some research [11], because smallholders, who are the main subjects of our study, mostly focus on satisfying their own consumption as the goal of agricultural production, and the induced effect of weather factors on the adoption of innovative seeds by farmers only occurs when the business goal shifts to the pursuit of market profits [61].

### 4.1.3. Robustness Test

We used three empirical methods to test the robustness of the relationship between credit constraint and the adoption of innovative seeds (Table 7). Firstly, we replaced the probit model with a logit model for regression analysis. Second, based on the psychological effect of self-protection, individuals may unconsciously seek the middle option that is more consistent with the majority perception when faced with attitude questions [62,63]. Therefore, in order to avoid extreme values from influencing our findings, we removed the sample of farmers who chose "1 = very difficult, 5 = very easy" in response to the question, "If you need 50,000 yuan for production turnover, how easy is it to borrow money?" Finally, considering that most elderly people do not have the ability to engage in agricultural production or business activities, and they are often not the implementers of household activities. Referring to the study of Li et al. [64], we performed a multivariate probit regression after excluding the sample of older people aged 60 years or older. As we can see from Table 7, the results are consistent with the baseline regression, both in terms of the significance of the regression coefficients and the sign of the coefficients, indicating that the estimation results of the model are robust.

**Table 7.** Analysis of the moderating effects of interlinked insurance and credit contract.

| Variables | First Round of Experiments | | | Second Round of Experiments | | |
|---|---|---|---|---|---|---|
| | Model Replacement | Excluding Extreme Values | Transformation Samples | Model Replacement | Excluding Extreme Values | Transformation Samples |
| Credit constraint | −0.8096 *** (0.12) | −0.6013 *** (0.12) | −0.191 *** (0.04) | −1.0125 *** (0.13) | −0.8725 *** (0.13) | −0.169 *** (0.03) |
| Interlinked insurance and credit contract | 1.0284 *** (0.26) | 0.4319 ** (0.17) | 0.304 *** (0.07) | 0.9833 *** (0.27) | 0.4585 ** (0.19) | 0.180 *** (0.06) |
| Weather | | | | −0.2406 (0.27) | −0.3090 (0.19) | −0.0264 (0.06) |
| Control Variables | | Controlled | | | Controlled | |
| Regional dummy variables | | Controlled | | | Controlled | |
| LR Chi2 | 132.69 | 79.00 | 91.14 | 164.21 | 115.68 | 91.14 |
| Prob > Chi2 | 0.0000 | 0.0000 | 0.0000 | 0.0000 | 0.0000 | 0.0000 |
| Pseudo $R^2$ | 0.2309 | 0.1989 | 0.2445 | 0.2871 | 0.2931 | 0.2499 |
| Log Likelihood | −221.0421 | −159.1246 | −141.2127 | −203.9015 | −139.5304 | −136.7986 |
| Number of samples | 415 | 288 | 275 | 415 | 288 | 275 |

Note: ***, ** indicate significance at the 1%, 5%statistical levels, respectively, and standard errors are in parentheses.

### 4.2. Moderating Effects of Interlinked Insurance and Credit Contract

From the theoretical analysis, it can be seen that interlinked insurance and credit contracts can mitigate the inhibitory effect of credit constraint on farmers' innovative seeds adoption behavior effectively. In order to analyze whether the provision of interlinked insurance and credit contract facilitates the weakening of credit constraints and increases

the adoption of innovative seeds by farmers. We used the group regression method to test the moderating effect of interlinked insurance and credit contract; in other words, we wanted to know the differences in credit constraint on farmers' innovative seed adoption behavior separately, taking into account whether interlinked insurance and credit contract are provided in the two rounds of experiments. The results are shown in Table 8.

**Table 8.** Moderating effects of interrelated insurance and credit contract.

| Variables | First Round of Experiments | | | | Second Round of Experiments | | | |
|---|---|---|---|---|---|---|---|---|
| | Not Available | | Available | | Not Available | | Available | |
| | Coefficient | Marginal | Coefficient | Marginal | Coefficient | Marginal | Coefficient | Marginal |
| Credit constraint | −0.3060 *** (0.08) | −0.1219 *** (0.03) | −0.8876 *** (0.14) | −0.3448 *** (0.06) | −0.4645 *** (0.09) | −0.1796 *** (0.04) | −0.8800 ** (0.14) | −0.3250 *** (0.06) |
| Weather | | | | | −0.1933 (0.21) | −0.0740 (0.08) | 0.0790 (0.29) | 0.0294 (0.11) |
| Control Variables | | Controlled | | | | Controlled | | |
| Regional dummy variables | | Controlled | | | | Controlled | | |
| LR Chi2 | 49.21 | | 109.69 | | 64.78 | | 121.86 | |
| Prob > Chi2 | 0.0000 | | 0.0000 | | 0.0000 | | 0.0000 | |
| Pseudo R$^2$ | 0.1591 | | 0.4223 | | 0.2086 | | 0.4704 | |
| Log Likelihood | −130.0886 | | −75.0201 | | −122.8414 | | −68.5982 | |
| Number of samples | 224 | | 191 | | 224 | | 191 | |
| Experience *p*-value | | 0.000 | | | | 0.014 | | |

Note: ***, ** indicate significance at the 1%, 5% statistical levels, respectively, and standard errors are in parentheses. The "empirical *p*-value" is used to test the significance of the difference in the coefficient of "credit constraint" between groups, which is obtained using a seemingly uncorrelated model test.

From the results, it is clear that credit constraint significantly and negatively affects farmers' innovative seeds adoption behavior in both rounds of the experiment, regardless of whether an interlinked insurance and credit contract is provided, indicating that credit constraint is a strong factor affecting farmers' innovative seeds adoption [65]. Moreover, we can also draw the conclusion that the inhibitory effect of credit constraint on farmers' adoption of innovative seeds is stronger when an interlinked insurance and credit contract is provided. Compared to no interlinked insurance and credit contract offered, farmers' adoption of innovative seeds is lower, which is inconsistent with hypothesis 3 proposed in this paper. This suggests that interlinked insurance and credit contracts not only fail to effectively transfer farmers' risk but also reinforce the inhibitory effect of credit constraint on farmers' adoption of innovative seeds. As Giné and Yang conclude, the emergence of the interconnected credit and insurance model may provide a danger signal for farmers, and the adoption of innovative seeds makes farmers' agricultural behaviors risky [19], which results in a decrease in farmers' acceptance of innovative seeds. Furthermore, the mitigation effect of interconnected insurance and credit contract on farmers' credit constraints does not directly contribute to their adoption of innovative seeds [66]. Therefore, the mechanisms of their influence need to be further explored.

### 4.3. Further Analysis of Moderating Effects

Farmers' acceptance of innovative technology is positively correlated with their knowledge [19,33,67], and a lack of knowledge about complex information, such as insurance, may enhance the complexity of the technology diffusion process [3]. As a result, for further analysis, we regress the sample farmers' mean value of years of education. According to the regression results (see Table 9), in both rounds of the experiment, the interlinked insurance and credit contract is designed to guide the more educated farmers to adopt innovative seeds. The finding that farmers' educational level is significantly beneficial to raising the adoption rate of innovative seeds is supported by numerous documents [68], but the mechanism of the effect needs to be further verified.

**Table 9.** Interlinked insurance and credit contract and farmers' adoption of innovative seeds.

| Variables | First Round | | Second Round | |
|---|---|---|---|---|
| | Education ≥ 7.5 | Education < 7.5 | Education ≥ 7.5 | Education < 7.5 |
| Interlinked insurance and credit contract | 0.170 ** (0.07) | 0.140 (0.09) | 0.102 * (0.06) | 0.0830 (0.09) |
| Observations | 247 | 168 | 247 | 168 |

Note: **, * indicate significance at the 5%, and 10% statistical levels, respectively, and standard errors are in parentheses.

Table 10 illustrates the regressions of the differences in farmers' adoption rates of innovative seeds regardless of the provision of interlinked insurance and credit contract. We can see that the educational level of households positively promotes farmers' acceptance rates if interlinked insurance and credit contract is not offered. Otherwise, the influence is not significant. Insufficient information and limited knowledge are generally considered to be the main factors that hinder the adoption of agricultural innovative seeds in general [50,69], while complex technology is usually knowledge-intensive and requires higher understanding abilities, so education may play a key role in promoting technology extension [67,70]. Interlinked insurance and credit contract, as a novel concept, is still in their infancy in China. Therefore, it is common sense that the education level of farmers is a key factor influencing their acceptance rate of innovative seeds.

**Table 10.** Differences in farmers' acceptance of interlinked insurance and credit contract.

| Variables | First round of Experiments (Marginal Effect) | | Second Round of Experiments (Marginal Effect) | |
|---|---|---|---|---|
| | (1) Treatment Group | (2) Control Group | (3) Treatment Group | (4) Control Group |
| Credit constraint | −0.345 *** (0.06) | −0.122 *** (0.03) | −0.325 *** (0.06) | −0.180 *** (0.04) |
| Education | −0.0147 (0.02) | 0.0379 *** (0.01) | −0.00122 (0.02) | 0.0281 ** (0.01) |
| Control Variables | Controlled | Controlled | Controlled | Controlled |
| Regional dummy variables | Controlled | Controlled | Controlled | Controlled |
| Number of samples | 191 | 224 | 191 | 224 |
| LR chi2 | 109.69 | 49.21 | 121.86 | 64.78 |
| Prob > Chi2 | 0.0000 | 0.0000 | 0.0000 | 0.0000 |
| Log likelihood | −75.0201 | −130.0886 | −68.5982 | −122.8414 |
| Pseudo $R^2$ | 0.4223 | 0.1591 | 0.4704 | 0.2086 |

Note: ***, ** indicate significance at the 1%, 5% statistical levels, respectively, and standard errors are in parentheses.

Credit will naturally flow to the activity with the highest marginal returns if the financial market is perfect [71]. However, the insurance and credit markets are still imperfect in China, and a complete separation of consumption and production decisions is not possible, especially in rural areas of China [71,72]. The demand for complex insurance in China is limited because farmers may be implicitly covered by limited liability in the contract that combines credit with insurance and charge premiums that actually raise the interest rate on loans [19,72].

Giné and Yang, Farrin, and Miranda provide the positive interlink among the degree of education, the adoption of innovative seeds, and willingness to loan. Although innovative seeds have higher yields, they also have greater risks [1,19,37]. The relatively low level of education of farmers in the sample area and their lack of awareness of the potential benefits and the potential risks associated with more complex financial products made it difficult to eliminate farmers' distrust, even though the interlinked insurance and credit contract was explained in detail prior to the beginning of the experiment. Therefore, reducing the

adoption of interlinked insurance and credit contract products becomes a reliable choice. The empirical study by Bewket concludes that the adoption and widespread diffusion of soil and water conservation technology are not sustainable for Ethiopian farmers [73], which is in line with our findings. According to Table 11, the average years of education of farmers in different groups are only 7.54 and 7.48; most farmers have not completed junior high school or higher education ($\geq$9 years), and the lower education level results in farmers' inability to make accurate assessments of complex insurance terms. Therefore, the demand for interlinked insurance and credit contract in China's agricultural development process remains highly limited [42], and thus improving farmers' education will mitigate the inhibitory effect of credit constraints on farmers' adoption of innovative seeds.

**Table 11.** Test for differences in means.

| Variables | Treatment Group | Control Group | Differences | P |
|---|---|---|---|---|
| Education | 7.54 | 7.48 | −0.06 | 0.8212 |
| Risk | 0.2128 | 0.2889 | 0.0761 ** | 0.0171 |

Note: ** indicate significant at the 5% statistical levels, respectively.

When comparing the farmers without an interlinked insurance and credit contract, their adoption of innovative seeds is 22% and 15% lower in the two rounds of experiments, respectively (see Table 8). It can be seen that farmers' acceptance rate of innovative seeds is higher in the second round of experiments than in the first round, which might be caused by the lag effect of farmers' adoption behavior of innovative seeds [67]. Gine et al. also argued that farmers' poverty might deem the information obtained from the initial experiments with a new technology of lower value and adopt it later [73].

Giné and Yang indicate that risk-averse borrowers prefer growing traditional varieties over adopting riskier hybrid varieties [19]. We compared the ambiguous risk preferences (Equation (9)) of the control and treatment groups. Table 11 shows that the mean value of ambiguous risk preferences of farmers in the control group is significantly higher than that of the treatment group, indicating that farmers in the treatment group are more risk-averse when they are borrowers. Therefore, the provision of interlinked insurance and credit contract will increase the acceptance of riskier hybrid varieties by risk-averse farmers.

## 5. Discussion

Based on data from the field experiment with 415 rural households in two Chinese provinces, this study explored the inhibitory effect of credit constraint on farmers' adoption of technology from the perspective of innovative seed technology. This study also explored the heterogeneity of rural households' adoption of innovative seeds with or without interlinked insurance and credit contract products. Compared with previous studies, the contributions of this study can be summarized as follows: (1) This study constructs a theoretical analysis and systematic framework of "credit constraint → interlinked insurance and credit contract → adoption of innovative seeds technology". (2) This study quantitatively explores the impact of credit constraints on farmers' adoption of innovative seeds. (3) This study further analyzed the factors that impede the demand for interlinked insurance and credit contract in current China's agricultural development using empirical evidence regarding the inhibitory effect of credit constraint on the adoption of agricultural technology. The study will also motivate policymakers to improve rural credit markets in order to alleviate farmers' credit constraints and thus promote the adoption rate of innovative seed technology. Last but not least, it will prompt government departments to provide diversified infrastructural support for innovative technology adoption, such as innovative seeds. Moreover, credit and insurance markets are boosting farmers' acceptance of innovative seeds.

Additionally, the results of our analysis show that credit constraint hinders farmers' adoption behavior of innovative seeds significantly. This result is consistent with the findings of Tesfaye et al. [43] and Boinec et al. [74], who indicated that reducing credit

constraints benefits agricultural production. Specifically, credit constraint is a common problem faced by farmers in most developing countries. There are significant information asymmetries in rural credit markets, which may lead to adverse selection and moral hazard problems. Since innovative seeds are high-risk, high-return investments that require financial support, credit becomes an inevitable way of agricultural investment for small-scale farmers. The research results show that using financial tools can significantly promote the adoption of innovative seed technology [32,49,50].

In addition, the relationship between the control and dependent variables in this study is mostly consistent with the findings of previous studies. For example, Adebayo et al. [10] conclude that farmers with more years of education have a better understanding and responsiveness to new agricultural technology, and they are more likely to have more positive attitudes toward new technology. Our study finds that farmers with more years of education are more likely to adopt innovative seed technology. Similar to the findings of Li et al. [63], we found that household income is an important driving force for the adoption of innovative seed technology. Only farmers with sufficient financial support can afford the cost and are more willing to adopt innovative seed technology. Coinciding with the findings of Tang et al. [6] and Giné et al. [19], we validated the role of risk preferences in promoting technology adoption by rural households. We found that technology training cannot promote farmers' technology adoption behavior, which is inconsistent with Mariano et al. [75]. According to Table 5, the majority of farmers received less than one training session in the previous year, indicating that systematic technological training services for farmers in the study area require improvement.

The results of this study also validate the factors that impede the diffusion of interlinked insurance and credit contract products in developing countries. We found that interlinked insurance and credit contract products may further reinforce the inhibiting effect of credit constraint on farmers' adoption behavior of innovative seeds, which is inconsistent with Qiu et al. [32], Farrin et al. [37], Carter et al. [40], Liu et al. [49] and Li et al., who [50] all point out that the "credit + insurance" partnership approach is an effective way to help farmers access more capital. However, this study found that rather than alleviating the inhibiting effect of credit constraint on farmers' adoption of innovative seed technology, the introduction of interlinked insurance and credit contract product reinforces the exacerbation. Giné and Yang [19] specify that the emergence of interlinked insurance and credit contract may provide a dangerous signal for farmers and that the adoption of innovative seeds makes farmers' agricultural behavior riskier, hindering farmers' choice of innovative technology. Bridle et al. [65] argued that this may also be caused by the fact that the mitigating effect of interlinked insurance and credit contract on farmers' credit constraint does not directly contribute to farmers' adoption of innovative seeds and that the mechanisms of their influence are not yet to be explored. Our results suggest that farmers' demand for interlinked insurance and credit contract in rural China is still limited at the current stage.

Further, this study proposes the effectiveness of improving farmers' education to mitigate the inhibitory effect of credit constraints on the adoption of innovative seeds from a cognitive perspective. Giné and Yang [19], Hörner et al. [66], and Oyawole et al. [66] all argued that farmers' knowledge comprehension significantly affects their acceptance of innovative technology. In the field of agricultural technology production, the lack of understanding of complex information may increase the complexity of the technology product diffusion process. The findings of this study are also in line with the fact that innovative seeds are knowledge-intensive and complex technologies, and farmers are not sufficiently aware of the potential benefits and the potential risks associated with credit-insured interactive products. At this point, rational farmers may choose to refuse the new technology. In addition to this, Gine et al. [58] and Hörner et al. [66] argued that compared with rich farmers, relatively poor farmers might generate a lower valuation of the information obtained from the initial experiments with the new technology and thus adopt it later. Similar to his findings, farmers' acceptance of innovation seeds is 22% and

15% lower in the two rounds of experiments when interlinked insurance and credit contract is offered to farmers compared with no interlinked insurance or credit contract (see Table 8). In this paper, the acceptance rate of innovative seeds by farmers in the second round of experiments is higher than that in the first round, verifying the validity of the delayed effect of farmers' innovative seeds adoption behavior.

The findings of this study are consistent with the fact that smallholder farmers are mostly risk-averse. In contrast, risk-averse borrowers may prefer growing traditional varieties to adopt riskier hybrid varieties. Therefore, the provision of interlinked insurance and credit contract should, in principle, increase the acceptance of risk-averse farmers and thus mitigate the inhibitory effect of credit constraints on the adoption of innovative seeds by farmers.

On the other hand, there are still some shortcomings in this study that can be improved further.

(1) This paper uses cross-sectional data to explore the relationship between credit constraint, interlinked insurance and credit contract, and innovation seeding technology adoption. However, the relationship among them may be dynamic. Therefore, future studies can apply panel data to fill the gap of the time-varying issue. (2) Due to the limitations of field experiment implementation, this paper assumes technology adoption as an innovative seed technology adoption behavior. However, agricultural technology behavior is divergent. In future research, we can design more scientific and detailed field experiments to study farmers' adoption behaviors of different innovative technologies in a categorical manner and improve the research system. (3) The sample area and sample size of this paper are still limited and unable to reflect extensive findings. In future studies, we can focus on the differences in technology adoption behaviors among different areas, including eastern, central, and western China, or southern and northern China, to draw more comprehensive conclusions.

## 6. Conclusions and Policy Implications

The low adoption rate of new agricultural technology is one of the major obstacles faced by small farmers in China. In fact, the Chinese government has been committed to the development of seed technology and has made many efforts to promote the application of agricultural technology. However, factors such as the credit constraint hindered this process. At the same time, there is widespread recognition of the role of interconnected credit and insurance partnership models in promoting the adoption of agricultural technology. However, innovative seed technology is costly and long-lasting, and farmers' willingness to adopt it varies depending on different regions and crops.

Overall, the findings suggest that credit constraint is a strong variable that hinders farmers from adopting innovative seed technology and that the link between insurance and credit contract has a significant positive effect on farmers' adoption of innovative seed technology. Other factors that drive farmers' innovative behavior in introducing varieties are complex and diverse, including their education level, risk preference level, family income level, government technological service support, weather disaster information services, and distance to the nearest financial institution. It is worth noting that this moderating effect further indicates that the interlinked insurance and credit contract will further enhance the inhibitory effect of credit constraint on farmers' innovative seeds adaption behavior. Currently, domestic demand for tying insurance and loan agreements is relatively low; at the same time, there is a lag in the introduction of innovative seeds by farmers.

Based on the empirical findings, this study provides valuable insights for policymakers to develop strategies to encourage farmers to adopt innovative seed technology sustainably. In this regard, first of all, it is necessary to further ease credit constraints and promote farmers' adoption of innovative seed technology. Specifically, the agricultural financial policy and credit interest rate term structure should be improved to effectively solve the problem of information asymmetry in the credit market (adverse selection and moral

hazard). At the same time, the cooperation space between rural insurance and credit institutions should be further expanded to promote the deepening of rural finance. The government adopts a more inclusive attitude, increases the innovation of agriculture-related credit products and services, expands the scope of mortgages and pledges of agriculture-related loans in accordance with laws and regulations, and increases credit support for key areas of agriculture. In addition, promoting the continuous improvement of farmers and financial literacy can also further ease credit constraints to a certain extent. Then, for banks and insurance institutions, it is necessary to strengthen benchmarking policy requirements and support China's three rural areas to make up for shortcomings.

Based on empirical results from winter wheat growers in northern China, this paper validates the effectiveness of promoting cooperative interaction between agricultural insurance and rural credit markets to facilitate innovative seed adoption promotion. The interlinked insurance and credit contract can effectively reduce farmers' uncertainties, providing a means to diversify agricultural investment risks while helping farmers solve their financial constraints and promoting the adoption of innovative seeds by farmers. Therefore, there is a need to further expand the scope of this model in China and enhance the linkage between the insurance market and the credit market.

In addition, government agencies need to provide basic support for the adoption of innovative technology, such as innovative seeds, through multiple channels. For example, the government can provide technological guidance for farmers to adopt innovative seeds through technical training and demonstrations and strengthen the publicity of natural disaster risks in agricultural production and meteorological disaster warnings to improve farmers' risk perception. It is evident that droughts and rainy days are bottlenecks to regional agricultural development, so it is necessary to strengthen publicity to help farmers avoid agricultural yield reduction caused by natural disasters and realize the transformation of agricultural development bottlenecks into potential agricultural development.

Finally, farmers' education levels need to be improved to enhance their understanding of interlinked insurance and credit contract products. Because the demand for interlinked insurance and credit contract is still relatively limited at the current stage of China's agricultural development, improving the education level of farmers can alleviate the inhibiting effect of credit constraints on farmers' adoption of innovative seeds. Increasing investment in rural education can improve the education level of farmers, thereby enhancing farmers' awareness of joint interlinked insurance and credit contracts and increasing farmers' acceptance rate of innovative seeds.

**Author Contributions:** Conceptualization, L.Y., Y.S. and H.W.; Methodology, L.Y. and Y.S.; Software, L.Y.; Validation, L.Y. and Y.S.; Formal analysis, H.W. and H.S.; Investigation, H.W. and H.S.; Resources, H.W. and H.S.; Data organization, L.Y., H.W. and H.S.; Writing—original draft preparation, L.Y. and Y.S.; Writing—review and editing, L.Y. and Y.S.; Supervision, H.W. and H.S.; Project management, H.W. and H.S.; Funding acquisition, H.W. and H.S. All authors have read and agreed to the published version of the manuscript.

**Funding:** This research was funded by National Natural Science Foundation of China General Program "Economic Incentive, Peer Effects and Green Fertilization Behavior of Wheat Growers: A Randomized Controlled Trail", grant number "71973087"; National Natural Science Foundation of China Youth Fund Project "Effects of Incentive Heterogeneity on Agricultural Technology Extension: A Randomized Controlled Experiment in Northern Wheat of China", grant number "72003215"; The 72nd general program of China Postdoctoral Science Foundation "Digital Agricultural Technology Extension and Farmers' Green Fertilization Technology Adoption: Mechanism of Action, Welfare Effect and Policy Optimization", grant number "2022M720170"; Soft Science Project of Science and Technology Department of Shaanxi Province "The impact of Incentive Heterogeneity on the Extension of Green Agricultural Technology: A case study of Wheat Water and Fertilizer Integration Technology in Weihe Plain", grant number "2022KRM131"; The Special Fund project of Basic Scientific Research Operation funds of Central Universities "The Influence of Peer Effect on Farmers' Green Fertilization Behavior: Based on the Analysis of the Randomized Controlled Experiment of Winter Wheat in Fen-Wei Plain", grant number "20SZYB21".

**Data Availability Statement:** The data presented in this study are available on request from the corresponding author. The data are not publicly available due to the data used in this study is derived from the project team's field research, which is highly private in nature.

**Conflicts of Interest:** No conflict of interest exists in the submission of this manuscript, and the manuscript is approved by all authors for publication. I would like to declare on behalf of my co-authors that the work described was original research that has not been published previously and is not under consideration for publication elsewhere, in whole or in part. All the authors listed have approved the manuscript that is enclosed.

## Appendix A. Research Questionnaire

1. Interviewee:
What is your gender? (1 = male, 0 = female);
How old are you ___; what is your nationality ___? (1 = Han, 0 = minority);
How many years have you attended school;
Are you a party member (1 = yes, 0 = no);
Are you a village cadre (1 = yes, 0 = no);
Are you a household head? (1 = yes, 0 = no)

2. Head of household information: gender ___ ; age ___; attended years of schooling ___; whether party members ___ (1 = yes, 0 = no); whether village cadres (1 = yes, 0 = no)

3. Does anyone in your family serve as a village official or civil servant?___(1 = Yes, 0 = No)

4. You have been engaged in agricultural production for ___ years, planting wheat for ___ years and corn for ___ years.

5. Your family has a population of ___ people (statistical caliber: 1. households at home, students with a collective account and military personnel), of which, the population older than 65 years old and younger than 16 years old are
Among the labor force, there are ___ people who work at home (including ___ male labor force and ___ female labor force), and there are ___ people who work outside all year round (working outside for more than 9 months a year).

6. In 2020, the income of your family from farming is ___ Yuan; the income from farming is ___ Yuan; the income from business is ___ Yuan; the income from cadres' salary is ___ Yuan; the income from agricultural subsidy is ___ Yuan; the income from farming is ___ Yuan.

7. How many miles from your home to the nearest river? ___ and how many miles from the nearest financial institution such as rural credit society ___.

8. Is there a weather information officer in your village? ___ (1 = Yes; 0 = No); Does the village provide technological information services to prevent meteorological disasters? ___ (1 = Yes; 0 = No)

9. What is the total area of wheat planted in your household in 2020? ___ (mu)

10. What is the number of blocks? ___

11. Does your family have family members or relatives working in financial institutions? ___ 1 = Yes, 0 = No

12. If you need 50,000 yuan for production turnover, how easy or difficult is it for you to borrow? ___
1 = very difficult; 2 = some difficulty; 3 = okay; 4 = easy; 5 = very easy

13. Did your household purchase wheat planting insurance last year? ___ 1 = yes; 0 = no.

## Appendix B. Experimental Design

### Risk Preference Experiment

1. First stage istest game: tell the tested farmer that the bag contains three black cards and three red cards, and the rewards for drawing black cards and red cards are shown in Table A1 as reward plan A1 and plan B1, respectively. The farmer's choice is ___(Note: Only those who choose plan B2 are allowed to continue the game).

**Table A1.** Test Games (Yuan).

| Risk Options | Plan A1 | | Plan B1 | |
|---|---|---|---|---|
| | Red card | Black card | Red card | Black card |
| | 15 | 20 | 16 | 21 |

2. The second stage is formal testing: After respondents have tried and become familiar with the rules of the experiment, the investigator provides 10 sets of test games; each set of test games includes two reward options: low risk and high risk, and respondents made risky choices for all 10 sets of games. Respondents selected either Reward plan A2 or Reward plan B2 from each of the 10 sets of games, with Reward plan A2 being the low-risk option and Reward plan B2 being the high-risk option. The second stage focused on making respondents understand that their choice of risky option is directly related to their final payoff to ensure that the information they displayed about their risk preferences is true and credible. In this stage, this study sets up two premises to measure the degree of risk preference for both deterministic and ambiguous probabilities, respectively.

(1) (See Table A2.) Respondents are explicitly informed that there are three black cards and three red cards in the bag: the number of respondents choosing plan B2 ___.

**Table A2.** Experimental protocol.

| Options | Plan A2 | | Plan B2 | |
|---|---|---|---|---|
| | Red Card | Black Card | Red Card | Black Card |
| 1 | 20 | 20 | 22 | 18 |
| 2 | 20 | 20 | 23 | 17 |
| 3 | 20 | 20 | 25 | 15 |
| 4 | 20 | 20 | 35 | 15 |
| 5 | 20 | 20 | 37 | 13 |
| 6 | 20 | 20 | 40 | 10 |
| 7 | 20 | 20 | 52 | 8 |
| 8 | 20 | 20 | 54 | 6 |
| 9 | 20 | 20 | 56 | 4 |
| 10 | 20 | 20 | 60 | 0 |

The first time a farmer jumps from plan A2 to plan B2 is option ___[number should be: 0–10]; let the farmer take any set of options for the actual experiment, and the farmer's reward is ___yuan [amount in the table * 0.1].

(2) (See Table A3) The respondents are explicitly informed that the bag contained six red and black cards of varying numbers and that only one color is known to have more cards. The number of respondents who choose plan B2 for the 10 sets of test games in Table A3 is ___.

**Table A3.** Experimental protocol.

| Options | Plan A3 | | Plan B3 | |
|---|---|---|---|---|
| | Red card | Black card | Red card | Black card |
| 1 | 20 | 20 | 22 | 18 |
| 2 | 20 | 20 | 23 | 17 |
| 3 | 20 | 20 | 25 | 15 |
| 4 | 20 | 20 | 35 | 15 |
| 5 | 20 | 20 | 37 | 13 |
| 6 | 20 | 20 | 40 | 10 |
| 7 | 20 | 20 | 52 | 8 |
| 8 | 20 | 20 | 54 | 6 |
| 9 | 20 | 20 | 56 | 4 |
| 10 | 20 | 20 | 60 | 0 |

The first time the farmer jumps from plan A3 to plan B3 is the option ___ [number should be: 0–10]; let the farmer take any set of plans for the actual experiment. The payoff to the farmer is ___ [amount in table * 0.1].

3. The experimental scenario of "Interlinked insurance and credit contract"

Situational Assumptions: In the baseline experiment, it is assumed that the farmers have an initial capital of CNY 4200 at the beginning of the experiment. They also have two types of wheat seeds available before the first year of cultivation: No.0 seeds and No. 1 seeds are available. The production inputs and income under different weather conditions differed between the two seeds (See Table A4: In the first year, farmers are required to make a choice of wheat seeds between No. 0 seeds and No. 1 seeds. The differences in seeds, production inputs and income under different weather conditions of these two kinds of seeds are shown in Table A5. If farmers choose No. 0 seeds, their own capital is enough and can satisfy the demand of production; if they choose No. 1 seeds, farmers need to apply for a loan of CNY 1800 and submit collateral worth CNY 1800 (we assume that the farmers can afford). At the end of the year, if the weather conditions are good for growing wheat (good weather), the farmers who plant No. 1 seeds can obtain the collateral back after repaying the loan. However, when the weather is bad, the farmers who plant No. 1 seeds cannot repay the loan because there is no income from planting, and the bank will confiscate the collateral. This means that they will end up losing CNY 1800, meaning that they have gone bankrupt, and the final payoff recorded by the experimenter is CNY −1800. Other farmers who are not bankrupt continue the experiment, and the final payoff in this experiment is determined by the amount of money left in each farmer's hand at the end of the last year. The experiment is conducted for a total of two years. The weather conditions for each year are determined in a similar way as in the risk experiment, with each farmer drawing a random card from a black box containing four red cards and two black cards, with a black card representing bad weather and a red card representing good weather. Weather conditions are simplified into two categories: disaster and normal. Based on local meteorological hazards, the incidence of disaster weather in the experiment is determined to be 1/3. Before the end of each round of the experiment, the farmers decide the weather conditions they would encounter in that round of the experiment by drawing lots. Farmers are required to select seeds at the beginning of the year: No. 0 seeds is low-cost, low-return that have been applied for many years; No. 1 seeds is innovative seeds with relatively high costs and returns, which are highly influenced by weather and have 1.55 times higher losses than normal seeds in case of bad weather. Farmers do not know what the weather will be like that year when they make the decision, so it is a risky decision. Farmers who purchase No. 1 seeds have to take a loan from the bank and provide collateral of corresponding value, which will be returned to the bank when the loan cannot be repaid. The main parameters involved in the experiment, such as production inputs and planting income, are shown in the table below. Both control and treatment group experiments are conducted in two rounds, representing two cropping cycles.

**Table A4.** Experimental parameter settings Comparison of traditional and innovative seeds (Unit: Yuan).

| Experimental Group | Seed Variety | Production Input | Loan | Premium | Planting Income (Good) | Balance (Good) | Planting Income (Bad) | Balance (Bad) |
|---|---|---|---|---|---|---|---|---|
| Control group | No. 0 seeds | 4000 | 0 | 0 | 7000 | 7200 | 0 | 200 |
| | No. 1 seeds | 6000 | 1800 | 0 | 12,000 | 10,200 | 0 | −1800 |
| Treatment group | No. 0 seeds | 4000 | 0 | 0 | 7000 | 7200 | 0 | 200 |
| | No. 1 seeds | 6000 | 2000 | 200 | 12,000 | 10,000 | 0 | 0 |

**Table A5.** Experimental parameter settings comparison of traditional and innovative seeds (Unit: Yuan).

| Experimental Group | Seed Variety | Production Input | Loan | Premium | Planting Income (Good) | Balance (Good) | Planting Income (Bad) | Balance (Bad) |
|---|---|---|---|---|---|---|---|---|
| Control group | No. 0 seeds | 4000 | 0 | 0 | 7000 | 7200 | 0 | 200 |
| | No. 1 seeds | 6000 | 1800 | 0 | 12,000 | 10,200 | 0 | −1800 |
| Treatment group | No. 0 seeds | 4000 | 0 | 0 | 7000 | 7200 | 0 | 200 |
| | No. 1 seeds | 6000 | 2000 | 200 | 12,000 | 10,000 | 0 | 0 |

**First round**

Step 1, farmer attributes ___ (control group = 0; treatment group = 1); farmer's technology choice ___ (traditional seeds = 0; innovative seeds = 1).

Step 2: The farmer draws lots to determine the weather conditions they will experience during the year ___ (0 = bad weather, 1 = good weather).

Step 3: The farmer's annual fund balance is calculated in dollars ___ [balance*0.001], and the farmer is informed of the current status.

After the first round of experiments, the farmer's choice ___ (0=quit agricultural production and go out to work, 1 = continue agricultural production). Note to farmer: if you choose 0, there is no need to do the second round of experiments; if you choose 1, please conduct the second round of the same experiment while keeping the attributes of the farmer unchanged.

**Second round**

Step 1: Farmers' technology choice ___ (traditional seeds = 0; innovative seeds = 1).

Step 2: Farmers conduct a lottery to determine the weather conditions they will experience that year ___ (0 = bad weather; 1 = good weather).

Step 3: The farmer's annual fund balance is calculated as ___ [balance*0.001] and the farmer is informed of the current status.

The overall amount obtained by the farmer from the experiment is ___ Yuan.

**In order to provide a global reference, we provide here the new experimental design parameters in USD.**

**(1 CNY = 0.14 USD).**

**Table A6.** Experimental parameter settings comparison of traditional and innovative seeds (Unit: USD).

| Experimental Group | Seed Variety | Production Input | Loan | Premium | Planting Income (Good) | Balance (Good) | Planting Income (Bad) | Balance (Bad) |
|---|---|---|---|---|---|---|---|---|
| Control group | No. 0 seeds | 560 | 0 | 0 | 980 | 1008 | 0 | 28 |
| | No. 1 seeds | 840 | 252 | 0 | 1680 | 1428 | 0 | −252 |
| Treatment group | No. 0 seeds | 560 | 0 | 0 | 980 | 1008 | 0 | 28 |
| | No. 1 seeds | 840 | 280 | 28 | 1680 | 1400 | 0 | 0 |

## Appendix C. Pictures of the Research

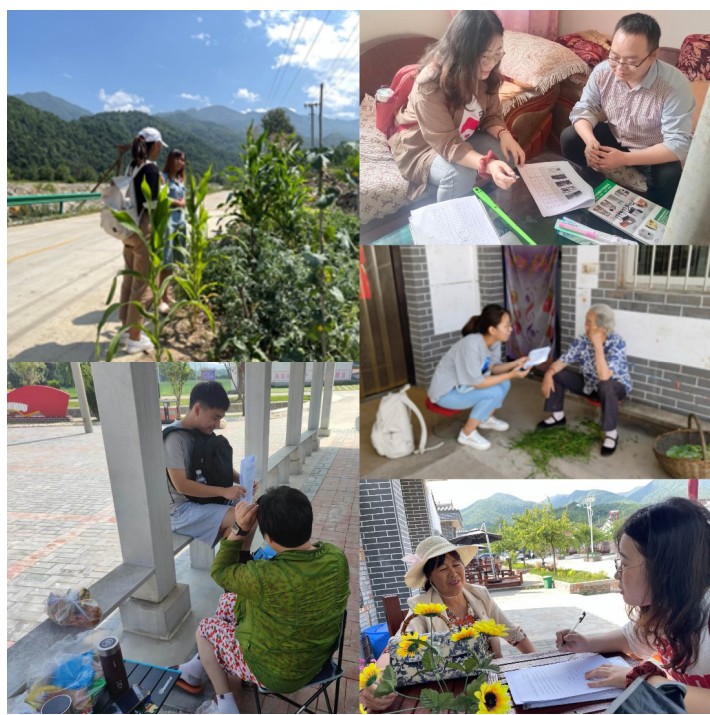

**Figure A1.** Data research and farmer interaction. Source: Photographed and compiled by the authors in July–August 2021.

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
