# Peer review of "Credit Constraint, Interlinked Insurance and Credit Contract and Farmers’ Adoption of Innovative Seeds-Field Experiment of the Loess Plateau"

_land, doi:10.3390/land12020357_

Round 1

Reviewer 1 Report (Previous Reviewer 1)

Thanks for accepting my comments and the improvement made in the manuscript. 

Author Response

Thank you very much for your positive comments and we are honored to receive your approval after revising the article. Based on the previous draft, we made further changes to the language of the article and added some additional illustrations.

Reviewer 2 Report (Previous Reviewer 2)

In light of the comments from the earlier review reports, the current manuscript has undergone extensive revisions. I thank the authors for the changes to the paper and their responses. I trust the manuscript is now ready for publication with a minor revision/check.

Author Response

Thank you very much for your positive comments. Certainly, your previous valuable suggestions in the previous draft are also greatly appreciated. Based on the previous draft, we made further changes to the language of the article and added some additional illustrations.

Reviewer 3 Report (Previous Reviewer 4)

No comments.

Author Response

Thank you very much for your previous valuable suggestions in the previous draft. Because of that, your suggestions have played an important role in enhancing the article. Based on the previous draft, we further revised the language expression of the article and added some additional illustrations.

Reviewer 4 Report (New Reviewer)

The study addresses an important issue. Innovation, especially high-yield and high-risk innovation, is characterised in many studies by high entry costs and the asymmetry between costs and benefits in the long run are often difficult for small farmers to bear. It is important to emphasise that this is not only the case in developing countries, where knowledge barriers are often very significant, but also in developed countries. 

Furthermore, considerable information asymmetries exist in rural credit markets worldwide, which certainly deserve attention. In this sense, this study can help to construct a solution. 

This study not only helps to argue that the use of financial instruments can significantly promote the adoption of innovative seeds, but also to provide a basis for the classification of companies that are more or less likely, due to their socio-structural characteristics, to innovate. 

The theoretical background is well constructed.

A review of the English language is necessary, and some synthesis work may be required at times. However, the paper is well structured.

Author Response

 Thank you for your kind comments on our article. Based on the previous draft, we have further revised the language expression of the article and added some additional illustrations. We also hope that our article will provide a useful reference for technology adoption by farmers in rural areas of developing countries.

Reviewer 5 Report (New Reviewer)

Dear authors,

This is an interesting article on credit and insurance

I have few suggestion and comments for further improvement

over all there needs a editing in the language, grammatical errors. do use simply shorter sentences to covey the message.

Rewrite line 84-91 and 121-122 and 267-268

lines 100-102 repeats in the manuscript.

It would be better to provide a graphical representation of the sampling procedure followed. And photographs or pictures of you data collection and farmer group interactions.  Also try to add location map of study regions.

clarify why was robustness test was done by doing other regression analyses. also in line 510 and 511 it is given probit was replaced by logit model regression. then why the analysis is not done ?

The references are adequate.

Thank you

Author Response

Thank you for your kind comments on our article. In response to your suggestions, we made changes and provided a point-to-point response here.

This manuscript is a resubmission of an earlier submission. The following is a list of the peer review reports and author responses from that submission.

Round 1

Reviewer 1 Report

My comments are attached in the file. 

Reviewer 2 Report

The research article is substantive and within the scope of the journal. The paper is generally well-written and well-researched, though it would benefit from an English language revision.

I have a few comments/suggestions for the authors:

1.     The abstract must indicate the background or motivation of the research. Need to also add the specific research method.

2.     The keywords should be specific and short.

3.     The survey questionnaire and experimental procedure should be attached in the appendix section.

4.     How do you confirm that the farmers understood the experimental procedure well? Do you have any validation questions in this regard?

5.     How realistic or real-time are the experimental parameter settings for numerical values like production input, loans, premiums, and planting income in the bed and good environment? To what extent do farmers disagree with these guidelines?

6.     Initial capital is 4200 Yuan or 4000 Yuan (Line #268)?

7.     Why is the number of cards (red and black) different (Line #286)? It implies that there is a greater likelihood of good weather, correct?

8.     Should add a reference to the following statement ‘Meanwhile, the loss will be 1.7 times higher than that of normal seeds if bad weather occurs’ (Line #291-292).

9.     What is the reasoning behind selecting Plan B to continue the game? It is unclear.

10.  The value of Plan A seems to have no risk (same value) in Table 4, yet in the text, it appears to have a low risk. Does it make sense?

11.  What rationale is used in Table 4 to define the maximum and minimum values of Plan B? How are these numbers produced? Make it clear.

12.  Discussion is an essential component of research that explains in depth how your research's theoretical implications link to science, practice, and policy. But I found that the discussion of the results was fragmented by the authors, making it exceedingly difficult for readers to express their understanding of the study clearly. Therefore, I would strongly recommend separating the discussion from the results section.

13.  Conclusions should be updated with the limitations of your study and future extensions.

Good Luck!

Reviewer 3 Report

Thank you for inviting me to review this manuscript. Although the topic is interesting but manuscript is poorly written. It needs significant improvement in order to make it publishable in a well reputed journal.

The authors selected specific variables to influence farmers’ adoption of innovative seeds. It was clear which variables would have a significant effect.

The text in the Conclusions is reiterating the results and discussions and has rather low contribution. Should be reformulated by clearly emphasizing the results and their impact for the academic community. Discussion part is shallow and no comparisions in the context of China.

Reviewer 4 Report

This is an interesting econometric or experimental economics paper with a particularly attractive overall experimental design. The experimental process of the field experiment restores the real process of agricultural production practices. In addition, the measurement of weather conditions and risk preferences in the paper is very interesting.

Of course, there are several areas where the authors could improve.

1. The moderating effect analysis section of the article comes to a surprising and unconvincing conclusion. The next part of the article has further analysis that this impediment stems from the lower education level of farmers. The interpretative perspective is reasonable and leads to meaningful conclusions that are highly original. However, can the authors examine this issue in terms of differences between regions? For example, Shaanxi is divided into different geographic regions, such as southern Shaanxi, central Shaanxi, and northern Shaanxi, do the different environments cause differences in the results?

2. The “Discussion” section or “Conclusion” section needs to be further strengthened. For example, in the first paragraph, it is mentioned that “Then for banks and insurance institutions, they need to strengthen their alignment with policy requirements and support China’s “three rural areas” to make up for the shortcomings. How to strengthen the link between them? The authors could give more focused conclusions. The third paragraph of the policy proposal, for example, is very clear. “For example, the government can provide technical guidance for farmers to adopt innovative seeds through technical training and demonstrations.”

 In the next paragraph, the author mentions that “improve farmers’ education level and enhance their knowledge of index insurance.” This paper focuses on the “Inter-linked insurance and credit contracts”, not on index insurance. There is a big difference between them. It is suggested to check and revise these irregularities, and other more meaningful discussions can also be conducted.

3. Finally, the overall idea of this paper is clear, but the relationship diagram between the key variables of the study is not presented throughout the paper. In a related study in this field, a diagram either complex or simple, as long as it is visual, could be used to present the relationship between the key variables of the article. In this way, it would be easier for the reader to read and understand. Because the article actually draws a paradox with the hypothesis, it is important and innovative to make the reader understand “the negative moderating effect of Inter-linked insurance and credit contracts in the process of credit constraints affecting the farmers’ adoption of innovative seeds.

Round 2

Reviewer 3 Report

I reviewed the article in the first stage and decided to reject it.